Food selection and feeding patterns in nectarivorous bats: Leptonycteris yerbabuenae and Glossophaga soricina

http://orcid.org/0000-0002-9608-2738 de Santiago-Hernández Martín Hesajim 1 2
Salinas-Melgoza Alejandro 1 2
Chavez-Estrada Alicia 1
http://orcid.org/0000-0003-3209-1659 Salinas-Melgoza Miguel Angel 3 4
Quesada Mauricio 2
http://orcid.org/0000-0001-5037-8878 Herrerías-Diego Yvonne 1 2 yvonne.herrerias@umich.mx
1 Laboratorio de Vida Silvestre/Facultad de Biología, Universidad Michoacana de San Nicolás de Hidalgo , Morelia, Michoacán de Ocampo , Mexico
2 Laboratorio Nacional de Análisis y Síntesis Ecológica, Escuela Nacional de Estudios Superiores, Unidad Morelia, Universidad Nacional Autónoma de México , Morelia, Michoacán de Ocampo , Mexico
3 Escuela Nacional de Estudios Superiores, Universidad Nacional Autónoma de México , Morelia, Michoacán , Mexico
4 Facultad de Biología, Universidad Michoacana de San Nicolás de Hidalgo , Morelia, Michoacán , Mexico
Brygadyrenko Viktor
Electronic publication date: 2025 Oct 24
Publication date: 2025
Volume: 13
Electronic Location ID: e20164
Received 2024 Dec 16; Accepted 2025 Sep 11
Copyright: © 2025 de Santiago-Hernández et al.
Copyright year: 2025
Copyright holder: de Santiago-Hernández et al.
License: This is an open access article distributed under the terms of the Creative Commons Attribution License, which permits unrestricted use, distribution, reproduction and adaptation in any medium and for any purpose provided that it is properly attributed. For attribution, the original author(s), title, publication source (PeerJ) and either DOI or URL of the article must be cited.
License URL: https://creativecommons.org/licenses/by/4.0/

Keywords: Foraging, Food preference, Nectarivory, Niche partitioning, Behavior

Funding: Universidad Nacional Autónoma de México PAPIIT IV200418, IN224920, IN219021, IN225924 SADER CONAHCyT 291, 333, CONAHCyT-National Repositories 271, 432, CONAHCyT-UNAM-UAGro-UMSNH Laboratorio Nacional de Análisis y Síntesis Ecológica LANASE 2015LN250996, 2016-LN271449, 2017-LN280505, 2018-LN293701, 2019-LN299033, 2020-LN314852, 2021-LN315810 LANASE-CIC-UNAM 2015–2024 Programa Iberoamericano de Ciencia y Tecnología para el Desarrollo RED CYTED-SEPODI 417RT0527 UMSNH CVU: 326281 This work was supported by grants from Universidad Nacional Autónoma de México (PAPIIT IV200418, IN224920, IN219021, IN225924), SADER-CONAHCyT 291, 333, CONAHCyT-National Repositories 271, 432, CONAHCyT-UNAM-UAGro-UMSNH to Laboratorio Nacional de Análisis y Síntesis Ecológica LANASE (2015LN250996, 2016-LN271449, 2017-LN280505, 2018-LN293701, 2019-LN299033, 2020-LN314852, 2021-LN315810), LANASE-CIC-UNAM 2015–2024, and Programa Iberoamericano de Ciencia y Tecnología para el Desarrollo RED CYTED-SEPODI (417RT0527). CONAHCyT Postdoctoral fellowship to Martín Hesajim de Santiago-Hernández (CVU: 326281) at UMSNH. The funders had no role in study design, data collection and analysis, decision to publish, or preparation of the manuscript.

==============================
Sympatric species reduce competition for resources due to differences in one or more of their niche dimensions. Biotic interactions between pollinators and variations in the availability and quality of resources are important factors that determine food selection in bats. The nectarivorous species Leptonycteris yerbabuenae and Glossophaga soricina coexist temporarily in much of their distribution and depend on nectar to feed. These species have similar requirements but differ in the way they obtain food. The coexistence of bat species with similar requirements, such as L. yerbabuenae and G. soricina, suggests that these species have strategies for avoiding competition and maximizing their nectar consumption. However, it is unclear how these bat species select resources and adjust their visits to the available floral resources. We therefore analyzed nectar selection and feeding patterns in these two bat species under captive conditions. We conducted experiments in which we controlled resource type and its availability by offering the bats different artificial nectar solutions, while we removed interspecific interactions. These solutions differed in concentration and sugar type, and some were similar to the nectar offered by chiropterophilic plant species. The bat species presented differences in food selection; G. soricina fed mainly on resources similar to Ipomoea and sucrose sugar. In contrast, L. yerbabuenae preferred those resources similar to the nectar of Acanthocereus cacti. In addition, the timing of feeding for each solution also differed. These results suggest low levels of competition between species under abundant resources and low density of individuals; however, such conditions are not always found in nature, and patterns may change with increased food scarcity and a high density of competitors.

Introduction

Coexistence between sympatric species is possible because species use resources differently. This phenomenon is known as niche partitioning (Schoener, 1974; Tavizon, 1998; Denzinger & Schnitzler, 2013; Salinas-Ramos et al., 2015). Niche partitioning can occur in two main dimensions: feeding time and diet niche flexibility (Schoener, 1974). In the new world, nectar-feeding vertebrates such as hummingbirds and phyllostomid bats are hovering animals that depend on constant nectar consumption to cover their high metabolic rates (Hainsworth & Wolf, 1972; Winter & Von Helversen, 1998; Winter, Voigt & Von Helversen, 1998). The search for floral resources results in high energetic costs for hovering animals because floral nectar differs between plant species in production rate, quality, quantity, composition, and spatial and temporal distribution (Ratnieks & Balfour, 2021). For this reason, hummingbirds and bats follow distinct strategies to obtain their feeding resources and avoid competition among nectarivorous sympatric species (Tschapka, 2004). These two nectarivorous groups exhibit differences in their daily activity patterns; hummingbirds forage primarily during the day, while bats are active at night. Both bats and hummingbirds use a trap-lining strategy to obtain nectar, but only hummingbirds exhibit physical interference to defend the flowers they feed on Fleming, Muchhala & Ornelas (2005). Therefore, bat species foraging during the night may present a niche partitioning strategy that can help them to coexist.

Nectar can vary in many traits, such as sugar concentration and sugar composition. Some experimental studies have not reported differences in bat preference for a given sugar type when in equal concentrations; however, results from these studies suggest that nectarivorous bats prefer highly sugar-concentrated nectars that represent the most energetic resources (Rodríguez-Peña et al., 2007; De Santiago-Hernández, 2013). Specifically, plants pollinated by bats may produce a volume of nectar ranging from 100 μl to 20 ml in one night (Tschapka & Dressler, 2002; Nicolson & Thornburg, 2007), and nectar sugar concentration can vary significantly from 3% to 33% (Baker, Baker & Hodges, 1998, Rodríguez-Peña et al., 2007). However, the most common sugar concentration ranges from 18% to 21% between chiropterophilous plant species (Von Helversen & Reyer, 1984; Baker, Baker & Hodges, 1998; Rodríguez-Peña et al., 2016). Sugar composition in the nectar of bat-pollinated plant species is primarily composed of glucose, fructose, and small amounts of sucrose (Wolff, 2006; Rodríguez-Peña et al., 2016). This constancy in nectar composition among chiropterophilous plant species may limit the dimensions into which niche partitioning may occur, as this may increase the likelihood of species preferences overlapping.

The coexistence of bats with similar food preferences may increase the likelihood of competition (Bloch, Stevens & Willig, 2011). Modeling competition studies in nectarivorous bats suggests that body size alone cannot describe niche differentiation and coexistence (Bloch, Stevens & Willig, 2011). Foraging behavior is an important component of niche differentiation and may promote coexistence between bat species that share food sources, have high energetic requirements, and have similar phenotypical adaptations (Novella-Fernandez et al., 2020; Muchhala et al., 2024). Thus, a specialized diet in a few specific food items or a generalized diet in most items may promote the coexistence of bats with similar requirements. Specialist bats may be more selective on particular resources when sharing foraging sources than generalist bat species, which may use most available resources. Therefore, in this study, we analyze the feeding preferences of two nectarivorous sympatric bat species that differ in their specialization degree and share floral resource consumption. We test for food resource preferences using both sugar solutions of a single sugar type and solutions that emulate the sugar concentration and composition of the nectar of plant species used by bats in natural conditions. We expect that bat species sharing floral resources and differing specialization degrees will exhibit distinct foraging strategies to coexist. These strategies will involve using different resources and adjusting the timing of foraging during the night.

Materials and Methods

Study species

Leptonycteris yerbabuenae is one of the largest nectarivorous bats in America (Cole & Wilson, 2006). This species can fly up to 100 km per night to reach feeding areas (Horner, Fleming & Sahey, 1998), and northern populations perform migrations during the winter, tracking flowering plants (Wilkinson & Fleming, 1996; Herrera Montalvo, 1997; Rojas-Martínez et al., 1999; Cole & Wilson, 2006; Morales-Garza et al., 2007). In contrast, Glossophaga soricina is a smaller nectarivorous bat (Alvarez et al., 1991) and has much more reduced mobility than the other species, flying up to three km per night in search of feeding areas (Aguiar, Bernard & Machado, 2014). Although both species of bats use nectar as the principal food resource, fruit and insect consumption may occur when nectar is partially unavailable (Howell, 1979; Rojas-Martínez et al., 2012). However, insect and fruit consumption is more common in G. soricina than in L. yerbabuenae (Gardner, 1977; Chávez-Estrada, Salinas-Melgoza & Herrerías-Diego, 2019).

Study site and bat sampling

Bats were collected in the “La Bonetera” area in Lázaro Cárdenas, southern Michoacán, Mexico (ca. 18°05′N, 102°25′W). This area is covered mainly by tropical deciduous forests and patches of semi-deciduous forests (Sandoval-Soto, 2013). It has a marked dry season from November to June and an average annual temperature of 27 °C (Cristóbal-Pérez, 2011). The area has reported six species of nectarivorous bats, with L. yerbabuenae and G. soricina as the most abundant species (M. A. Sandoval-Soto, 2013, personal observations). We used mist nets at ground level to capture 6 non-reproductive adult males of each L. yerbabuenae and G. soricina (Chávez-Estrada, Salinas-Melgoza & Herrerías-Diego, 2019). At the end of the study, all individuals were returned to their natural conditions. We followed Mexican laws for animal care, use, and handling (SEMARNAT permit no. SGPA/DGVS/03702/17) to YHD.

Care and housing of bats

All bats were transferred to the housing facilities in Morelia, Mexico, where they were marked with plastic collars, and their weight was constantly monitored. We kept bats inside 0.75 × 0.75 × 0.75 m cages, covered with shade cloth, in groups of three individuals. The place where the experiment was performed remained in darkness at night and with little illumination during the day. The temperature was maintained constant at around 26–27 °C, and relative humidity was around 50%; these conditions were similar to the capture area conditions and used in other experimental studies (Rodríguez-Peña et al., 2013). Bats were captive for 28 days, eight for climatization to experimental conditions, 12 days for experimental trials, and eight days after experimental trials to finally return to natural conditions. Every day, bats were fed at 20:00 h. with a maintenance diet composed of milk powder, cereals, sugar, and fruit (mango or banana), complemented with vitamins and minerals (multivitamin tonic, “Cariño”, Mexico) (Mirón et al., 2006).

Experimental design

We prepared two different types of sugar solutions (Table 1). First, we used three sugar solutions based on a single type of sugar, and second, we used sugar solutions emulating the nectar of three chiropterophilous plants used by the two bat species during the moment of their maximum abundance on the site where bats were captured (Chávez-Estrada, Salinas-Melgoza & Herrerías-Diego, 2019). For solutions with a single sugar type, we use glucose and fructose hexoses and the disaccharide sucrose in a concentration of 20%. To prepare sugar solutions that emulate the nectar of plant species with distinct compositions and concentrations of sugars, we used the sugar types in the proportions reported by Rodríguez-Peña et al. (2016, Table 1). For Ipomoea ampullacea (Convolvulaceae) (24.42 ± 1.03 °Brix), Ceiba aesculifolia (Malvaceae) (18.03 ± 0.93 °Brix), and Acanthocereus occidentalis (Cactaceae) (27.13 ± 1.44 °Brix). Additionally, we used purified commercial water as a control. To prepare sugar solutions at distinct sugar concentrations, we used a weight/weight concentration, which means that in a solution of 100 g at 20% sugar concentration, 20% of the solution weight corresponds to the sugar type and 80% to the water weight. We based the sugar solution preparation on weight/weight concentration because it offers easy preparation, precision, and easy measurement. Additionally, the use of sugar solutions based on weight/weight concentration allows comparison with previous studies (Rodríguez-Peña et al., 2007). We tested the sugar concentration of all solutions with a manual refractometer. Combining these solutions provided a gradient of sugar concentrations from 0 (pure water) to 27 °Brix, from which both species could choose.

Table 1 Composition of artificial nectar solutions provided to the bats during each experiment.

Sugar solutions used in experimental trials. Three solutions emulating the sugar concentration and sugar composition of the nectar of chiropterophilous plant species. Three solutions composed of only one sugar type; water was used as a control. Sugar solutions were based on the study of Rodríguez-Peña et al. (2016).

Solution	Plant family	Nectar concentration
(°Brix)	% Glucose	% Fructose	% Sucrose	Energy content in 100 g of solution (kcal)	
Acanthocereus occidentalis	Cactaceae	27.13	17.06	28.97	53.98	105.39	
Ipomoea ampullacea	Convolvulaceae	24.42	40.93	46.58	14.49	95.18	
Ceiba aesculifolia	Bombacaceae	16.85	47.18	49.82	2.99	65.33	
Fructose	–	20	–	100	–	80.00	
Sucrose	–	20	–	–	100	77.4	
Glucose	–	20	100	–	–	75.00	
Water	–	–	–	–	–	0.00	

To evaluate food selection, we selected six individuals per bat species. Each individual was exposed to one experimental night trial; therefore, we conducted 12 trials. Every night, we placed one individual in the experimental area, which consisted of a 3.6 × 1.6 × 2.6 m cage with two resource patches placed 1 m off the ground on opposite sides. We used 15-ml Falcon plastic tubes with 17 mm diameter as feeders. These feeders allowed bat snouts to enter, display their tongues, and feed on the sugar solution. Each patch had 14 feeders, two feeders for each experimental solution. Each feeder was placed randomly within the patch and filled with 15 ml of sugar solution one time per experimental trial. Bat’s behavior was recorded at night using two night-vision video cameras positioned at each food patch. In each experimental trial, recording began at 20:00 h. and ended at 07:00 am. for a total of 132 h of video recordings. We discarded the first recording hour for analysis since the bats took around an hour to feed after the experimental solutions were placed in the cage. The video recordings were reviewed in slow motion using the software VLC-media-player 2.2.6. To evaluate foraging behavior and activity patterns, we recorded each time the bats approached a feeder and drank from it. We also recorded the feeder they visited and the time each feeding event occurred. The data is available in File S1.

Data analysis

To determine whether the number of visits differs both between sugar solutions and per hour of foraging time, we use generalized linear models based on the Genmod procedure implemented in the SAS 2003 statistical software (ver. 9.4). Statistical tests were conducted separately at two different datasets: (a) each bat species and (b) data of both bat species together. We used the number of visits as a dependent variable and the sugar solution as an independent variable to test whether the number of visits of each bat species differs among sugar solutions. For the analysis with data of both bat species together, we used the number of visits as a dependent variable and the sugar solution and bat species as independent variables. To analyze whether the number of visits differs across foraging activity hours, we considered the total number of visits per bat individual per hour from 21:00 to 07:00 h. as a dependent variable and the hour as an independent variable. For the dataset of both species, we used the number of visits per hour as the dependent variable and the hour and bat species as the independent variable. We used a Poisson distribution with an associated log-link function and the ILINK function to back-transform data to the original scale for all statistical analyses. We calculated a type 3 effects analysis and a post hoc Tukey-Kramer test.

To analyze whether bat species differ in the selectivity for sugar solutions, we calculated the standardized specialization index Kullback-Leibler (d′) based on the Shannon diversity index that compares the interaction distribution of each bat species with each sugar solution using an interaction matrix of bat species with sugar solutions (Kullback & Leibler, 1951; Blüthgen & Menzel, 2006). Originally, the d′ values range from 0 to 1, where values close to one indicate maximum specialization; however, for our purposes, d′ indicated the selectiveness of bat species for sugar solutions. To determine whether the observed d′ value differs from those expected by chance, we compared it with values obtained from the r2dtable null model, which keeps the matrix sum and row/column sum constant (Dormann, Gruber & Fründ, 2008). We used 1,000 permutations with the r2dtable null model function implemented in the “bipartite” R package (Dormann, Gruber & Fründ, 2008; R Core Team, 2016). We consider the d′-index value significant when the empirical metric is larger than 95% of the null model’s results. This indicates that the empirical d′ significantly deviated from null expectations. Calculations for specialization d′ index were performed in the “bipartite” package ver. 2.18 implemented in R-software platform ver. 4.3.1.

Results

In the experiments, G. soricina individuals presented more feeding events during the night, with an average of 102 visits per night. In contrast, L. yerbabuenae individuals presented an average of 65 visits per night.

Significant differences were found between the number of visits to each sugar solution per bat species. L. yerbabuenae visited the feeders with nectar similar to Acanthocereus cacti more frequently (x2 = 259.48, df = 6, p < 0.0001) (Fig. 1). At the same time, G. soricina presented a higher number of visits to feeders with a solution similar to Ipomoea ampullacea nectar (x2 = 446.31, df = 6, p < 0.0001) (Fig. 1). When we analyzed for differences between the number of visits of the two bat species to sugar solutions, there were significant differences between the number of visits among sugar solutions by bat species (x2 = 348.06, df = 6, p < 0.0001). We observed that each bat species recorded a low number of visits to the solution preferred by the other species (Fig. 1). The water and the Ceiba aesculifolia solutions received the lowest number of visits by both bat species (Fig. 1). Only the water solution showed significant differences, with more visits by G. soricina. The fructose solution showed no significant differences in the number of visits between bat species.

Figure 1 Least square means of bat visits of experimental sugar solutions.

Least square means (Ls means) the number of bat visits per sugar solution. The top figures show the results of the analysis per bat species. The bottom figure displays the combined results of the analysis for the two bat species. Strong color bars indicate results for G. soricina, and light color bars indicate results for L. yerbabuenae. Different colors indicate each sugar solution. The two types of sugar solutions (those that emulate plant nectar and those with only one sugar type) are ordered from left to right in decreasing order of energy content. Error bars indicate standard error. Different letters indicate significant differences.

Bat species not only fed from different solutions, but their feeding activity also differed in distinct hours at night (L. yerbabuenae: x2 = 100.67, df = 10, p < 0.0001, and G. soricina x2 = 131.99, df = 10, p < 0.0001). Comparison between bat species indicates significant differences in activity at night (x2 = 56.73, df = 10, p < 0.0001), mainly due to L. yerbabuenae presenting two feeding peaks at 21:00–23:00 and 03:00 am, while G. soricina presented the highest number of feeding events at midnight (Fig. 2).

Figure 2 Visitation per hour of bat species.

Least square means (Ls means) of bat visits per hour. The top figures show the results of the analysis per bat species. The bottom figure displays the combined results for the two bat species. Error bars indicate standard error. Different letters indicate significant differences.

The number of visits to each solution varied throughout the night. Bats were recorded in the first hour of experimental trial apparently making some recognition visits to the feeders and then they chose the resources used for the rest of the night (Fig. 3). In the case of L. yerbabuenae, the consumption of water was restricted to the first hours of feeding only and was not consumed during the rest of the night (Fig. 3). In contrast, G. soricina consumed water throughout the night, although the number of visits was low and similar to Ceiba (Fig. 3). During the first hours of the night, both bat species visited various sugar solutions. Later, they concentrated their visits on particular sugar solutions. L. yerbabuenae from 23:00 h. visits were principally on a sugar solution similar to the cacti Acanthocereus occidentalis nectar. G soricina, after midnight, mainly consumed Ipomoea and sucrose solutions (Fig. 3). Finally, the selectiveness d′ index was higher for L. yerbabuenae (d’ = 0.19) than G. soricina (d′ = 0.12), indicating that L. yerbabuenae has more selectivity for sugar solutions than G. soricina. This result suggests that observed selectiveness is not derived from chance and is a result of other factors linked to the natural history of bats, in this case, to bats’ capacity to identify resources to feed and establish food preferences.

Figure 3 Alluvial plot showing the number of visits by bats to each sugar solution.

Foraging activity of bat species in response to different sugar solutions at various night hours. (A) Leptonycteris yerbabuenae (Ly) is represented in blue, and (B) Glossophaga soricina (Gs) is depicted in green. The left side of each figure illustrates the number of visits made by each bat species to the sugar solutions, while the right side shows the number of visits to each solution at different night hours. The middle boxes display the sugar solutions arranged in alphabetical order and categorized by type of sugar solution. The top three boxes represent sugar solutions that emulate the nectar of plants, followed by three solutions based on a single sugar type, and finally the water solution. The height of the boxes for bat species indicates the total number of visits made by each species to the sugar solutions. The height of the boxes representing sugar solutions reflects the number of visits received from the bat species, and the height of the hour boxes indicates the number of visits made by the bat species during each hour of the night. The thickness of the lines corresponds to the number of visits, and the colors of the links represent each type of sugar solution.

Discussion

Our results support our initial hypothesis of niche partitioning as a mechanism of coexistence of both L. yerbabuenae and G. soricina, based on the preferences for sugar concentration, sugar type, and the feeding time patterns. We found differences in resource use between bat species over time and in foraging behavior. Glossophaga soricina was more generalist than Leptonycteris yerbabuenae, while the latter species was selective and chose the most energetic resources. One possible explanation is that resource characteristics, such as nectar quality, could drive the feeding pattern (González-Terrazas et al., 2012), as indicated by the differences in the selectiveness d′ index. Concerning the activity patterns, we observed that bat species presented differences in their activity peaks: L. yerbabuenae peaked at 21:00–23:00 and 3:00 am, and G. soricina at midnight (Fig. 2). Our results also indicate that the duration of the feeding time per bat species was extended by 2 h more than previously reported by the study of Chávez-Estrada, Salinas-Melgoza & Herrerías-Diego (2019), ending at 7:00 am. The increase in bat feeding time occurred within the anthesis time of plant species such as Ipomoea ampullacea (19:00–07:00) and Ceiba pentandra (20:00 to 06:00), whose nectar sugar composition was emulated here (De Santiago-Hernández et al., 2019; Dzul-Cauich et al., 2025). Differences in nectar-type preference may serve as a strategy to reduce competition by enabling sympatric species to utilize the same floral resources in distinct ways.

Patterns of resource utilization

Our experiments show significant differences in the number of feeding events, feeding preferences, and feeding time according to bat species. The specialization d’ index suggests that individuals of Leptonycteris yerbabuenae were more selective in food resources than Glossophaga soricina. Differences in food preferences among nectarivorous species can be attributed to various factors, including physiological requirements (Horner, Fleming & Sahey, 1998; Ayala-Berdon, Schondube & Stoner, 2009), morphological specializations (Freeman, 1995; González-Terrazas et al., 2012; Bechler, Steiner & Tschapka, 2024), resource availability (De Santiago-Hernández, 2013), and nectar characteristics such as taste promoted by the presence of amino acids and other secondary metabolites (Nicolson & Thornburg, 2007; Ayala-Berdon et al., 2011; Rodríguez-Peña et al., 2021). Explanations accounting for this pattern may include both bat species’ intrinsic characteristics and resource characteristics. Some experimental studies have demonstrated that morphological specialization, such as the length of the oral apparatus and the tongue, is positively related to nectar extraction efficiency (González-Terrazas et al., 2012; Bechler, Steiner & Tschapka, 2024). The specialist species L. yerbabuenae is known to be more efficient at extracting nectar than the less specialized nectarivorous G. soricina (González-Terrazas et al., 2012). Therefore, G. soricina may conduct more visits than L. yerbabuenae to extract more sugar solutions and cover their energetic requirements.

Floral visits can also be influenced by nectar properties, such as viscosity, which increases with nectar concentration, making consumption more difficult (Baker, 1975; Kingsolver & Daniel, 1983; Nicolson & Thornburg, 2007). Experimental studies such as those of Bechler, Steiner & Tschapka (2024) and González-Terrazas et al. (2012) suggest that specialized bat species are more efficient in nectar extraction than generalists. So, the preference of the specialist L. yerbabuenae for the most concentrated and viscous nectar solution (A. occidentalis) may be related to this efficiency. For the less specialized G. soricina, the use of the second, more concentrated solution that emulates the I. ampullacea nectar may represent a resource with more energetic benefits than investing energy in obtaining viscous nectar. Unfortunately, the visitation data in this study do not allow for testing whether specialized bats have more efficiency than generalists in extracting viscous nectar, and whether the energy invested to obtain viscous nectar differs according to bat specialization. Therefore, future studies can analyze the relationship between nectar properties, such as viscosity, and bat specialization to improve our knowledge about the floral resource preferences of bats.

The energetic content and sugar composition of sugar solutions are another factor that may determine the bat’s preferences. For single sugar type solutions at 20 °Brix, results indicate that G. soricina visited the sucrose solution more than the fructose and glucose solutions (Fig. 1). Considering the caloric content of these solutions, sucrose represents the intermediate solution in energetic terms, containing 77.4 Kcal/100 g of solution, while fructose and glucose solutions contained 80 Kcal and 75 Kcal per 100 g of solution, respectively. On the other hand, G. soricina preferred the I. ampullacea solution over other solutions that emulate plant nectar and represent the second sugar solution with the most energetic content (95.18 Kcal) and sugar concentration (24.42 °Brix). Both sugar solutions contain a sucrose concentration of around 18% previously reported as the sucrose concentration preferred by G. soricina (Rodríguez-Peña et al., 2007, 2016) (Table 1). This result suggests that sucrose concentration may influence the nectar preferences of G. soricina, because the less visited nectar solutions were the A. occidentalis with high sucrose content (53.98%) and C. aesculifolia with low sucrose content (3.99%).

In contrast to G. soricina, L. yerbabuenae for single sugar type solutions, visited both glucose and fructose solutions equally; these sugar solutions have lower energetic content than the sucrose solution (Fig. 1). For sugar solutions that emulate the nectar of plants, the number of visits by L. yerbabuenae decreases as the sugar concentration and energy content reduces between sugar solutions (A. occidentalis (24.42 °Brix, 105.39 Kcal); I. ampullacea (24.42 °Brix, 95.18 Kcal); and C. aesculifolia (16.85 °Brix, 65.33 Kcal)). This order of selectiveness of resources suggests that the preferences of L. yerbabuenae are related to the energetic content of sugar solutions. Therefore, it is possible that L. yerbabuenae, one of the largest nectarivorous bat species, feeds on the most sugar-concentrated solutions to enhance their daily energy budget with less frequent feeding events (Von Helversen & Reyer, 1984; González-Terrazas et al., 2012).

Nectar, in addition to having distinct sugar types, contains amino acids and secondary metabolites that modify the taste and affect the abilities of nectarivorous bats to detect differences in sugar concentrations. For example, the study of Rodríguez-Peña et al. (2013) provides experimental evidence that L. yerbabuenae prefers solutions with high sugar concentrations (>25 °brix), similar to the A. occidentalis cacti nectar, as our study also shows. This study suggests that the presence of nitrogen and free amino acids in sugar solutions affects the ability of L. yerbabuenae to discriminate between sugar solutions with distinct concentrations but increases its preference for sugar solutions with these organic elements over those without them. In the case of G. soricina, the presence of amino acids and nitrogen in sugar solutions did not affect their preferences. Since our results are only based on bats’ preferences for solutions with distinct sugar concentrations and do not consider other nectar characteristics such as amino acid presence, taste, viscosity, or floral morphology and the flower position on plants, it is necessary to consider that bat preferences shown in this study may vary from the patterns found in nature.

Resource availability is another factor related to food selection in bats (Ayala-Berdon, Schondube & Stoner, 2009; Laurindo, Gregorin & Castro, 2017). Previous studies in tropical dry forests have found that Bombacaceous plant species constitute one of the most important resources for nectarivorous bats (Stoner et al., 2003; Quesada et al., 2003). Since experimental individuals were collected in a tropical dry forest, we expected they would feed mainly on these solution types, based on potential previous experience with these feeding resources. However, the sugar solution that emulated the nectar of C. aesculifolia was one of the least visited resources in our experiments. One possible explanation is that bats use Bombacaceous plant species in the wild because they have massive flowering that provides a higher volume of nectar than other plant species. For example, in a previous study in the study area, Pseudobombax ellipticum was the most consumed plant species by L. yerbabuenae and G. soricina during its massive flowering (Chávez-Estrada, Salinas-Melgoza & Herrerías-Diego, 2019). Nevertheless, our results in an experimental setting indicate that both bat species do not prefer Bombacaceous nectar. Then, the consumption of Bombacaceous nectar by the two bat species may be due to the large availability of flowers during mass flowering events rather than the quality of nectar.

Feeding time patterns

The timing of feeding activity also differed between species. The peaks of maximum visitation observed in our experiments indicate that G. soricina performed maximum visitation at 00:00 h., while L. yerbabuenae had two peaks at 21:00–23:00 h. and 03:00 a.m. (Fig. 2). These peaks of visitation occurred similarly to previously reported field data, which describes that for G. soricina, the maximum activity peak occurs at midnight, and for L. yerbabuenae at 22:00 h. and 03:00 AM (Chávez-Estrada, Salinas-Melgoza & Herrerías-Diego, 2019). Possibly, these peaks may be related to moments of higher nectar production during the night (Horner, Fleming & Sahey, 1998). However, in our experiments, nectar was available for bats throughout the night. So that maximum visitation may be related to the energetic requirements of each bat species (Fragaszy, Visalberghi & Fedigan, 2004) or the experience of bat species that were captured directly from the field. Bat field experience means that due to uncertainty in resource availability in the tropical dry forest, experimental individuals may have become accustomed to this and overlooked the consistent availability of food resources in the experimental setup. These results suggest that differences not only in food resource selection but also in the timing of peak activity can be niche partitioning strategies that allow them to optimize their food preference and modify their foraging behavior (Tschapka, 2004; Fleming, Muchhala & Ornelas, 2005). On the other hand, a limitation of our study is that we filled feeders only once at the beginning of the night, and they were not refilled thereafter. Therefore, it remains unclear whether changes in sugar solution volume among feeders due to bat consumption influenced bat preferences over the night. Thus, experimental and field tests that include the nectar consumption rate are needed to consider more aspects that may affect the preferences and the coexistence of bats with similar resource requirements.

Conclusions

Our results support our initial hypothesis that bat species that share floral resource consumption and differ in specialization have distinct nectar preferences and feeding time patterns to obtain floral resources and coexist. However, many other factors can influence foraging behavior under natural conditions, including biotic interactions, food quantity and quality, and physiological demands (Ayala-Berdon, Schondube & Stoner, 2009; Laurindo, Gregorin & Castro, 2017). Since our experiments were carried out under conditions of high resource abundance and low density of individuals, we suggest future experiments in which natural conditions are considered.

Supplemental Information

Supplemental Information 1 Raw data of visitation bats.

Visitation bats used for statistical analysis, including visitation data for each experimental treatment and the weight of each individual used in experimental sessions.

Supplemental Information 2 Arrive checklist.

We also want to thank Keith MacMillan for helping during language editing services.

Additional Information and Declarations

Competing Interests

The authors declare that they have no competing interests.

Author Contributions

Martín Hesajim de Santiago-Hernández conceived and designed the experiments, analyzed the data, prepared figures and/or tables, authored or reviewed drafts of the article, and approved the final draft.

Alejandro Salinas-Melgoza conceived and designed the experiments, authored or reviewed drafts of the article, and approved the final draft.

Alicia Chavez-Estrada performed the experiments, analyzed the data, prepared figures and/or tables, authored or reviewed drafts of the article, and approved the final draft.

Miguel Angel Salinas-Melgoza analyzed the data, authored or reviewed drafts of the article, and approved the final draft.

Mauricio Quesada conceived and designed the experiments, authored or reviewed drafts of the article, and approved the final draft.

Yvonne Herrerías-Diego conceived and designed the experiments, analyzed the data, authored or reviewed drafts of the article, and approved the final draft.

Animal Ethics

The following information was supplied relating to ethical approvals (i.e., approving body and any reference numbers):

Secretaría de Medio Ambiente y Recursos Naturales provided full approval for this research (SGPA/DGVS/03702/17).

Field Study Permissions

The following information was supplied relating to field study approvals (i.e., approving body and any reference numbers):

Secretaría de Medio Ambiente y Recursos Naturales provided full approval for this research (SGPA/DGVS/03702/17).

Data Availability

The following information was supplied regarding data availability:

The raw data is available in the Supplemental Files.

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
