# Peer review of "Food selection and feeding patterns in nectarivorous bats: Leptonycteris yerbabuenae and Glossophaga soricina"

_PeerJ, doi:10.7717/peerj.20164_

## Round 0.1 · original submission · Major Revisions

Dear authors, I ask you to listen to the very useful comments of the reviewers. I also recommend making the article more informative. In the figures (except for Figure 3), I recommend displaying the data as a box analysis (median, first and third quartiles, minimum and maximum values), marking the letters above the boxes according to the results of the Tukey test or other method of multiple sample comparison.

·

Basic reporting

The study has some major flaws that need to be addressed. The major problem of the manuscript arise already in its abstract in the statement "Previous field studies have not reported competition for resources when evaluating the diet of these species; however, it remains unclear how competition is involved or whether this segregation is based on resource characteristics." Why was this research performed, if there is no reason, based on previous research, to think there exists a competition between the two studied species? This should be clearly stated.

Experimental design

The concentration gradient of the sugar solutions is actually only very small and due to the not quite interpretable results (e.g. the preference of Glossophaga soricina for sucrose, but at the same time a negative preference for a sugar mixture similar to cactus nectar, with the highest sucrose content) it would be much more interesting to give bats sweet solutions of much greater gradient.

Validity of the findings

The authors evaluate differences between species based on the number of visits to feeders, but do not analyze the volume of food consumed at all. After all, this could nicely explain the differences in activity - I would expect that, in short, Leptonycteris visited feeders less often, but consumed much more food per night. It is quite clear from the methodology that the authors precisely monitored the volume of food consumed and this data should be included in the manuscript! Differences in overall activity could also be explained by differences in the context of metabolic scaling – in general, smaller species have higher unit metabolism and thus higher energy requirements, and typically have to spend more time foraging.

Some of the conclusions and arguments used in the discussion are based on assumption that there is no difference between the feeding behavior of bats between the situation in nature and in the laboratory. At the same time, it is clear from many existing studies that nectar-eating bats do not search for food only based on energy it provides, but also based on taste, based on position and structure of inflorescence from which they consume nectar, and, last but not least, based on specific smell A solution of three types of sugars in a feeder is simply not the same as a flower in nature. I would appreciate it if this fact was taken into account in the discussion.

Could the authors explain why Glossophaga soricina in the experiment visited the feeder with the sucrose solution most frequently (bats obviously liked it) and at the same time the solution with the highest sucrose content, imitating the composition of sugars in cactus nectar, was visited rarely? What if the final taste of the solution of sugars, instead of its energetic content, the mantra of many studies focused on diet of nectar-feeding bats, is more important? Although could not be extracted from the results of the study, but it could be mentioned as a hypothesis to be tested in the future studies.

Additional comments

Lines 108-109: Is the sentence „Care and housing of bats, experimental design, and data analysis were prepared 109 prior to initiating the study.“ important? I suggest releasing it.

Lines 75-79: I suggest adding your expectations (hypotheses tested) to the the final part of the Introduction.
Line 184: I suggest moving the first sentence to the Materials and methods section
Line 186: Which confidence (CI) intervals you used (70% CI, 90% CI, 95 % CI…?). Please, specify this.

Lines 189-193: If I understand well the statistical test used, its significance gives only evidence that visitation to different solutions was non-random, but it is not really testing which solution was visited more and which was visited less. This could be inferred only from inspection of mean values (e.g. from Fig.1). Am I wrong about that? If yes than you should specify in the Methods section how exactly you performed inter-sample comparisons.

Lines 198-201: I have same feeling here as mentioned above. The significant tests just show that the frequency of visitation differed among different times but do not tell anything about when the peak feeding activity occurred. This information arose not from the statistical significance but from visual inspection of graph (Fig.2). Hence, the proper wording that that gives a sense would be, e.g. : The bat species not only fed from different solutions but also their feeding activity was nonrandomly distributed over the night time (L. yerbabuenae:x2= 68.16, df=11, p<0.0001; G. soricina:x2= 98.46, df= 11, p<0.0001 ). While L. yerbabuenae presented a feeding peak at 03:00 AM, G. soricina presented its highest number of feeding events around midnight (Fig.2)
Line 199: use 3:00 AM instead of 3:00 hours

Line 218: There is no clear indication what is your initial hypothesis in the Introduction.
Line 227-229: Is there a possibility that longer feeding time in the laboratory was due to ad libitum offer over the whole night-length? In natural conditions, many (or most?) chiropterophilous plants pump nectar to flowers immediately after dusk and as night progresses flowers wither quite soon. Consequently, the time available for feeding might be shorter under natural conditions. Could you cite some study focused on that topic?
Lines 241-251: Well, it is possible that differences in oral morphology may be related to extraction efficiency and, consequently, to frequency of flower visitation under natural conditions. But why it explains the differences observed in laboratory, where bats had ad libitum access to sugar solutions, which availability and extractability was the same?

Figure 1: What values denote whiskers? SDs or SEs or what? They look extremely similar or same – is that real? What statistical procedure did you use for comparisons of frequency of visitation between different solutions? If I understand well, the test you used just indicate there is non-random distribution of visits but does not provide comparisons between samples (solutions).
Figure 2: The test criteria given in the caption do not refer to differences between the two species, they only show that the distribution of activity is uneven over the night.

Reviewer 2 ·

Basic reporting

I think that the biggest issue (although minor!) the article faces is clarifying some of the processes around the experimental design and analysis. I think that the context is there in the introduction and the discussion does a good job of bringing in further context for interpreting the differences found between the target species. I would like to see more clarity and context particularly when it comes to the sugar set up (why choose weight/weight ratios instead of molarity, or caloric equality - for instance). Additionally, just a bit of clarifying and streamlining the language regarding the 3 sugars that mimic natural sources as well as 3 sugars that are 'pure' solutions. It was a just little unclear to me on my first read through exactly what had been done until I got through the discussion and figures.

Further, I think a bit more information is required about the set up of the generalized linear models and a differentiation between significance testing and model selection. It seems as though chi squared tests have been done to identify significant differences across sugar types and feeding timing within the model built for one bat, and also for the same sugar between both bats? But it is unclear to me exactly what tests were performed on which sets of data. This is (hopefully) largely an issue that can be adressed by clarifying the writing and not indicative of a confused analysis.

Experimental design

I think that the design is quite cool! I understand the limitations of live animal studies and think this level of data collection for 6 individuals of these 2 species allows for a realistic comparison (at least of these two populations, whether these populations are representative of either species as a whole is beyond the scope of this paper). My main concern, as addressed above, is that there is a lack of clarity regarding the analysis that I think should be straightforward enough to amend.

Validity of the findings

The only 'finding' I would recommend the authors think about is whether the specialization index (d) tells us that L. yerbabuenae is more specialized than G. soricina or whether it only tells us that they are, independent of each other, more or less specialized than chance would predict. The language is a little ambiguous in the text regarding how they use this statistic. Barring a couple of the clarifications I've advocated for, I think the study is valid. The authors do not over-claim or misrepresent what they found. They have designed an experiment that, while limited in scope, provides for a clear interpretation and is adequately set up and linked to the research goals claimed.

Additional comments

Figure 1 has letters on each of the bars with no mention of what they mean in the figure caption. Please address this. Figure 3 is really visually overwhelming and virtually uninterpretable (to me at least) in terms of what it is trying to represent. I think a clearer alternative would be 2 plots, one for each species, with a timeline on the x axis and the y axis representing the frequency of visits for a time window or cumulative visits through the night with different lines or bars indicating the various sugar solutions. And for figure 4, as I understand d' it is a point estimate and not a cummulative counted value and so a bar chart is inappropriate - in fact I think a figure is a little superfluous for this.

·

Basic reporting

The manuscript presents interesting research on nectar preference in bats but requires substantial revision to clarify the research question, improve statistical analysis, and refine the discussion.

Clarity and Language:

The manuscript presents relevant research on nectar selection and feeding behavior in Leptonycteris yerbabuenae and Glossophaga soricina. However, the text requires significant improvements in fluency, coherence, and English language use. Some sections are difficult to follow due to unclear phrasing and awkward sentence structure. I suggest that the authors revise the manuscript for clarity and seek language editing assistance.

Examples where language can be improved:
Line 60: the phrase 'may be' is unnecessary here, as the statement is supported by citations. Consider using 'is' instead to make it more definitive.
The connection between the high energetic costs of flight and nectarivory in the introduction (lines 40-42) is unclear. Consider rephrasing to highlight the specific constraints imposed by nectar as a resource.
The discussion has some abrupt transitions between ideas, making it difficult to follow the argumentation.

Introduction and Background

The introduction lacks a clear and concise statement of the research hypothesis or predictions. While the text discusses niche partitioning and nectarivory, it does not explicitly state the research question and how the study addresses it.
I recommend restructuring the introduction to move from general background information to specific research objectives. A clear research gap should be identified, followed by an explicit research question and hypothesis.

Figures and Tables:

The figures and tables contain useful data but require improvements in clarity:
Figure 1: It is unclear what the error bars represent. The caption should specify whether they indicate standard deviation or confidence intervals. Additionally, the significance of the different letter labels is not explained.
Figure 2: Please clarify what the error bars represent. Are they standard deviations, confidence intervals, or another measure of variability? Including this information in the figure legend would improve clarity. The y-axis label 'LS-Mean of frequency' is unclear. Could you specify what LS-Mean refers to and how it was calculated?
Figure 3: The alluvial plot is difficult to interpret in the connections between solutions and time due to the high number of overlapping lines. An alternative visualization might better illustrate the feeding patterns.
Figure 4: The specialization index results could be presented in text rather than as a figure, it does not seem necessary.

Experimental design

Research Question and Scope

The study is within the scope of PeerJ and addresses an interesting ecological question. However, the research question should be more clearly defined. The introduction mentions the importance of sugar type preference, but the discussion focuses more on solution type (flower species) rather than sugar composition. The authors should explicitly link their hypotheses and results to sugar type preferences, especially considering that Ipomoea nectar contains lower sucrose concentrations than other solutions tested.

Methods and Reproducibility

The experimental design is well thought out, but some methodological details need clarification:
Bat Acclimatization: The text does not specify how long the bats were housed before experiments or if there was an acclimatization period. This is crucial information for interpreting their behavior. Also, it is not specified the total time of captivity.
Feeder Dimensions: The description of the feeders should include dimensions (line 139).
The authors mention some methodological details for calculating nectar extraction (lines 145-147), but this data is not presented in the results or discussed in the manuscript.
The purpose of using control feeders is unclear, as the manuscript does not present nectar volume data or discuss the observed effects of evaporation on the solutions.
Was the nectar in the tubes refilled during the experiment? If so, this could have influenced visitation patterns. If it was not refilled, please mention this explicitly and discuss how nectar depletion might have affected the results.

Statistical Analyses:

The manuscript does not describe the parameters used in the null model analysis (lines 173-181) for specialization indices. It should specify key details commonly reported in the literature, such as the number of null models generated, the constraints applied (e.g., fixed row/column sums), the algorithm used to randomize the interaction matrix, and the statistical approach for assessing significance. Providing this information will enhance the transparency and reproducibility of the analysis.
It is unclear whether interspecies differences in total visit frequency were statistically tested (lines 186-188). Overlapping confidence intervals suggest that differences may not be significant, and statistical tests should confirm this.
The energetic content of the nectar solutions should be incorporated into statistical analyses rather than treating them as categorical variables. This would allow for a more quantitative assessment of preference.
It is unclear when you are referring to frequency of visits versus number of visits. To avoid confusion, consider explicitly defining both terms in the Methods section and ensuring consistent usage throughout the text. For example, in Figure 1, the figure title mentions 'average frequency of visits per treatment,' while the y-axis label states 'mean number of visits per treatment.' Clarifying and standardizing these terms will improve readability and accuracy.

Validity of the findings

Data and Statistical Rigor

The raw data are supplied, but some results are not fully supported by statistical evidence:
The manuscript states that feeding duration was longer than in previous studies (lines 227-229), but no data or figures support this claim. If feeding duration was measured, it should be explicitly reported. There may be confusion between the terms 'feeding duration' and 'number of visits,' as they represent different aspects of foraging behavior. Clarifying the distinction between these variables would help improve the interpretation of the results.
The discussion states that L. yerbabuenae preferred more energetic solutions (lines264-267), but no direct statistical test compares feeding frequency with energy content. The authors could run an analysis where visits are compared across solutions according to their energetic content, rather than discrete solution types.

Interpretation of Results

The discussion presents various potential explanations for the observed feeding patterns. However, it would benefit from a stronger connection to the data, focusing on evaluating which factors are most supported by the results rather than listing all possible influences on nectar preference.
Additionally, some comparisons with the literature lack specificity. For example, the text states that feeding peaks align with previous field studies (lines 298-302) but does not provide exact numerical values for comparison.

---

## Round 0.2 · Major Revisions

Dear Dr. Herrerías-Diego, I ask you to carefully correct the shortcomings pointed out by the reviewers and hope that the new version of this article can be approved for publication.

Reviewer 2 ·

Basic reporting

Overall the experiment is clearly communicated and I think the discussion is engaging and interesting. I’m glad to see this version is quite improved from the previous version. The authors clearly set a hypothesis, they cache it in the relevant literauture in the field, and they test that hypothesis adequately. My major concern in terms of clarity, fortunately, is just some grammar issues throughout the manuscript and some typos. Nothing that prevented my understanding of the work as a whole, but they did stick out to me. I’ll list some below, but this is not necessarily exhaustive. The introduction could use some paragraph breaks (may be an issue with conversion to PDF). The only other issue I have is the Alluvial plot. I still strongly feel that breaking the alluvial plot into two figures, one for each species, would greatly improve comparison and visibility. The other figures are much improved and readable.


Line 53 and 59, two sentences here start with “then,” which is not quite used appropriately. In this context I would opt for ‘for this reason’ for the sentence on line 53 and ‘therefore’ for the sentence starting on line 59
Line 87 The sentence that starts ‘Foraging behavior is an…’ should probably be split into two sentences
Line 202-203 has a section of text repeated twice, likely a hold over from an edit.
Line 206 You are determining whether the experimental d’ values differ from those expected by chance
Line 210 probably go with ‘we consider the d’ value significant if the metrics were larger…’
Line 231 should use ‘fed’ not ‘feed’
Line 243 This phrasing is a little confusing. Do you mean that early in the night nectar sources were more varied and that later in the night they were more consistent?
Line 255 I would use ‘partitioning’ not ‘segregation’

Experimental design

I would very much like to see the null expectation of d' reported not just a p value. Beyond that, I have no issues.

Validity of the findings

No comment

Additional comments

I think with these minor revisions, the manuscript will be a great addition to the literature on nectar feeding bats, foraging preferences, and niche partitioning.

·

Basic reporting

The manuscript is overall more polished and easier to follow than the previous version. The writing has improved, but there are still several sentences that feel disconnected or difficult to follow, especially in the introduction and discussion.

Introduction and Background
I consider that the introduction still lacks a clear and concise research question or hypothesis and a clear thread of the introduction leading to the hypothesis

Consider rephrasing the paragraph on nectar composition to better link it to your research. One possible structure: “Nectar can vary in many traits, such as sugar concentration and sugar composition. Some studies have explored how bats respond to varying concentrations (cite), but the role of sugar type preference at equal concentrations remains largely unexplored.”

Figures and Tables
Figures and tables are improved compared to the previous version. They now present data more clearly and effectively.

Table 1: Describe the table following the same order in which the data appear. Note that I. ampullacea appears twice, please clarify if this is an error. Also clarify that the "composition" column refers only to the solutions emulating plant species, not all solutions.
Figure 1: Much improved, more legible and intuitive.
Figure 3: "Intensity of visits" is used for the first time here; consider rephrasing to be more accurate or better defined.

Experimental design

Clarify whether your hypothesis relates to foraging preferences, sugar composition, or both. You mention at the end of the introduction "using different resources", I think it would be better to be more specific.

Methods and Reproducibility
The methods section is clearer and more detailed than before. Good job on the improvements.

Line 149: Specify that the sugar solutions differed not only in concentration but also in sugar-type proportions. Cite Table 1.

Line 169: Consider discussing how the lack of refilling the tubes could have influenced the bats' visitation patterns. Could bats have changed preferences if some tubes appeared emptier? Mention this as a potential methodological limitation.

Validity of the findings

Data and Statistical Rigor
The statistical results are generally well presented.

Line 228: Note that there was no significant difference in fructose preference between the species.

Line 235: According to the figure, there’s no significant difference between 21:00–23:00. Consider referring to this as a time range rather than only “22:00.”

Line 239: The figure does not clearly show that the bats use different resources throughout the night. Please clarify or revise the statement.

Interpretation of Results
The discussion section still needs work. It often lacks clear connections between statements and supporting data

Line 250: Add the biological relevance of this result.

Line 256: Clarify the phrase here. For example: "i.e., sugar type."

Line 260: Add something like "(as shown by the differences in d' index)" to connect with your results.

Line 264: Be more specific about how similar.

Line 267: Add full citation.

Line 271: The logic of the paragraph is unclear. Consider rephrasing to something like: “Differences in nectar-type preference may serve as a strategy to reduce competition by allowing sympatric species to exploit the same floral resources differently.”

Line 278: Consider adding citations after each factor mentioned in the list.

Line 285: This sentence is too general and doesn’t contribute meaningfully, consider removing it.

Line 297: Not sure this is supported by the literature. Add a citation or remove.

Line 299: There’s confusion between sugar concentration and caloric content. The preferred solution is intermediate in energy but not in concentration (all had the same concentration). This makes the reference to viscosity inaccurate as well. Revise this part.

Line 309: Add citation to Table 1 here.

Line 311: This statement may be valid for the pure sugar solutions but not for the flower-based ones, since those varied in both sugar composition and concentration. Also, A. occidentalis, which had high sucrose, was not preferred by G. soricina. Reconsider or clarify this.

Line 314: Rephrase to avoid awkward phrasing like “our experiments’ most concentrated and energetic resources.”

Line 351: Consider ending this paragraph with a sentence that directly ties back to your findings, rather than the current one, which leaves the idea hanging.

Line 353: You cite only one study, consider adding more to support your point, or change “Some studies”

Line 361 / Line 363: Be more specific, how similar or different?

Line 367: There’s no logical connection to the previous sentence. Add a transition.

Line 373–374: Not clear what “field experience of individual subjects” means or how it's supported by the data.

Line 377: Consider rewording. As written, it sounds like your results are about hummingbird foraging.

Line 385: Be more specific. Instead of saying “abilities,” say you tested “nectar preferences and foraging patterns.” Also, I would not call those “abilities”

Line 386: Consider changing the term, you could use the same subheading used in the discussion (“Feeding time patterns”) for consistency.

Additional comments

Overall, the manuscript is much improved from the earlier version. The methods are more clearly described, and the figures are more effective. The main areas needing improvement are the introduction, which needs a better structure and a clear hypothesis, and the discussion, where logical flow, connections to the data, and accurate interpretation need refinement.

---

## Round 0.3 · accepted · Accept

Dear Dr. Herrerías-Diego, I congratulate you on the acceptance of this article for publication.

·

Basic reporting

The manuscript is now much clearer and more polished. The introduction effectively threads the hypothesis. Figures and tables are significantly improved, particularly the replacement of Figure 3, which is now more intuitive and informative.

Experimental design

The research question and hypothesis are clearly defined, and the methods are detailed.

Validity of the findings

The discussion has been substantially revised for clarity, logical flow, and connection to the literature.

Additional comments

This manuscript makes a valuable contribution to the field. The authors have done an excellent job addressing the review comments and substantially improving the manuscript. The writing is clearer, the figures are more effective, and the introduction and discussion now present a stronger, more coherent narrative.